Epidemiology and resistant profile of bacterial pathogens in a tertiary health care hospital, Medan City: a retrospective study

Mayasari Evita 1 evita@usu.ac.id
Utama Edhie Djohan 2
1 Department of Microbiology, Faculty of Medicine, Universitas Sumatera Utara , Medan, North Sumatra , Indonesia
2 Siloam Hospital Medan , Medan, North Sumatra , Indonesia
Oliveira Sonia
Electronic publication date: 2025 May 27
Publication date: 2025
Volume: 13
Electronic Location ID: e19510
Received 2024 Dec 19; Accepted 2025 May 1
Copyright: © 2025 Mayasari and Utama
Copyright year: 2025
Copyright holder: Mayasari and Utama
License: This is an open access article distributed under the terms of the Creative Commons Attribution License, which permits unrestricted use, distribution, reproduction and adaptation in any medium and for any purpose provided that it is properly attributed. For attribution, the original author(s), title, publication source (PeerJ) and either DOI or URL of the article must be cited.
License URL: https://creativecommons.org/licenses/by/4.0/

Keywords: Bacteria, Identification, Antimicrobial agents, Susceptibility test, Prevalence, Multidrug-resistant organism, Polymicrobial isolates

Funding: The authors received no funding for this work.

==============================
Background

The epidemiology study of bacterial isolates and their resistance patterns in clinical settings is essential due to the rising threat of antibiotic resistance, which complicates treatment options. Understanding these patterns enables healthcare providers to tailor antibiotic therapies effectively, ensuring better patient outcomes and mitigating the spread of resistant strains. This study aims to analyze the prevalence and antimicrobial resistance patterns of microbes recovered from blood, wound, sputum, and urine specimens in a tertiary healthcare hospital.

Methods

In this retrospective cross-sectional study, we analyzed the clinical microbiology laboratory data of patients of all age groups from January 2022 to December 2023. Microbial isolates were identified using the commercial system bioMérieux-Vitek 2. Antimicrobial susceptibility tests were conducted using the Vitek 2 automated susceptibility system and disk diffusion methods, following the Clinical & Laboratory Standards Institute (CLSI) guidelines.

Results

A total of 1,190 (47.58%) microbial isolates were recovered from 2,501 clinical specimens, consisting of 98/714 (13.73%), 454/655 (69.32%), 394/545 (72.29%), and 244/587 (41.57%) from blood, wound, sputum, and urine, respectively. Escherichia coli was the most prevalent isolate identified from blood [27/98 (27.55%)], wound [107/454 (23.57%)], and urine [107/244 (43.85%)]. Sputum isolates were dominated by K. pneumoniae [68/394 (17.26%)]. Coagulase-negative staphylococci (CoNS) and S. aureus were the dominant Gram positives in all specimens. Polymicrobial isolates were found in 4/98 (4.08%) blood, 41/454 (9.03%) wound, 52/394 (13.2%) sputum, and 9/244 (3.69%) urine. The predominant polymicrobial pairs were K. pneumoniae and S. aureus [6/106 (5.7%)]. Poor sensitivity against cefoxitin and oxacillin was highest among the Gram-positives, dominated by CoNS. In Gram-negatives, sensitivity against ampicillin was the lowest.

Conclusions

A periodical update of the epidemiological profile of microbial isolates in hospital settings presented in this study is crucial for updating the empirical antibiotics and developing the prevention and infection control program.

Introduction

The global burden of infectious diseases remains a significant challenge, especially in low- and middle-income countries (LMICs), where social and economic vulnerabilities worsen health outcomes. The world is witnessing a resurgence of infectious diseases, exacerbated by the COVID-19 pandemic, which has resulted in an increase in other diseases (He et al., 2023). The global estimated mortality rate from infectious diseases per 100,000 people was 101.8 in 2019, but after the COVID-19 pandemic, it increased to 184.6 in 2021 (Institute for Health Metrics and Evaluation (IHME), 2024). Infections cause approximately 27.7% of the global disease burden, with bacteria accounting for 14% (IHME Pathogen Core Group, 2024). Sub-Saharan Africa has the highest estimated pathogen-associated burden in terms of disability-adjusted life years (DALYs) from all causes, while the high-income region has the lowest (IHME Pathogen Core Group, 2024).

Antimicrobial resistance (AMR) is estimated to have caused 1.91 million deaths worldwide, with an additional 8.22 million deaths predicted by 2050 (GBD 2021 AMR Collaborators, 2024). South Asia, Latin America, and the Caribbean are projected to have the highest AMR mortality rates in 2050 (GBD 2021 AMR Collaborators, 2024). The use of irrational and inappropriate antimicrobials in patients with infectious diseases is at risk of causing microbial resistance, and this occurs almost all over the world (Mbanga, Sithabile & Silence, 2015; Khan et al., 2023; Rajput, Johri & Goyal, 2023; Hossain et al., 2024; McGuinness et al., 2024; Sheeba, Prathyusha & Anila, 2024). The development of bacterial resistance to antimicrobials can reduce or lose the effectiveness of infectious disease treatment and cause morbidity, length of treatment, and patient mortality (Hunter et al., 2008). Antibiotic resistance can quickly transmit between bacteria and can even cause the emergence of pathogenic strains that are resistant to all antibiotics (Hunter et al., 2008; Pfeifer, Cullik & Witte, 2010; Flaifel, 2023; Kamel et al., 2024). Infection by those strains will be hazardous for patients, as there are no antibiotics that can be used for their management.

In May 2024, the World Health Organization (WHO) announced a list of priority infection-causing bacteria (World Health Organization (WHO), 2024). Bacterial pathogens classified as high-priority and critical pose a significant disease burden (mortality and morbidity), with increasing resistance trends, difficulty to prevent, and high transmissibility (World Health Organization (WHO), 2024). The ability to detect these pathogens needs to be improved to overcome infectious disease problems so that diagnosis can be followed by proper management. Susceptibility tests to antimicrobials in conjunction with microbial identification from clinical samples are crucial before selecting antimicrobials for infectious disease management. Susceptibility test data in hospitals are also vital to controlling antimicrobial use.

Despite a decrease in infectious disease burden between 1990 and 2019, communicable diseases such as tuberculosis, diarrheal diseases, and lower respiratory infections continued to be the leading cause of DALYs in Indonesia (GBD 2019 Indonesia Subnational Collaborators, 2022). Furthermore, irrational and inappropriate antimicrobial use complicated Indonesia’s infectious disease problems (Hadi et al., 2013; Endraswari et al., 2022). A study in Surabaya found that 60% of antibiotic prescriptions in hospitals were inappropriate, and 81% of 781 inpatients had multidrug-resistant (MDR) Escherichia coli (Hadi et al., 2013). We investigated the prevalence and antimicrobial resistance patterns of microbes isolated from blood, wound, sputum, and urine specimens at a tertiary healthcare facility in Medan City, North Sumatra. Epidemiological studies can help to monitor the prevalence of the pathogens and to understand their distribution and patterns of resistant strains, which is crucial for public health management.

Materials and Methods

Study design and ethics statement

This is a retrospective cross-sectional study conducted on patients’ medical records from a clinical microbiology laboratory in a private hospital in Medan, North Sumatra Province, Indonesia. Among the inclusion criteria for the data collection were: (1) all patients from all age groups who were admitted to the hospital from January 2022 to December 2023; (2) patients suspected of having an infection and having the microbiological analysis results of clinical specimens taken from the suspected focus of infection (blood, wound, sputum, and/or urine); and (3) patients admitted to the emergency department, wards (inpatient), and the hospital clinics (outpatient).

Clinical specimens other than blood, wound, sputum, and urine were excluded. Repeated cultures with microbial growth (positive) results from one patient for less than a 48-h interval were also excluded. This study was approved by the Ethics Committee of the Universitas Sumatera Utara in agreement with the Nuremberg Code and Declaration of Helsinki, as registered in letter number 1000/KEPK/USU/2023. This study does not require informed consent as it used anonymous patients’ medical records. The hospital’s ethics committee approved the consent waiver.

Collection of clinical specimens

Specimen collection and transport were processed according to the hospital protocol by following the literature (Church, 2016). At least 10 mL of blood was drawn from each adult patient. Blood sample volumes from neonates and pediatric patients complied with the body weight rule (Church, 2016). Pus, exudates, and swabs from wounds were categorized as wound specimens. Urine specimens were mostly collected by clean-voided midstream or urinary catheter methods. Urine specimens collected through suprapubic needle aspiration directly into the bladder were also accepted. Gram staining was used to determine the number of squamous epithelial cells (SEC) per low-power field (LPF) in wound (especially swab), sputum, and urine specimens that were eligible for culture. Sputum specimens containing 25 polymorphonuclear neutrophils and <10 SEC per LPF were accepted for culture.

Microbial isolation, identification, and antimicrobial susceptibility tests

Specimen culture for the isolations of bacteria and yeast was done according to the protocol of processing, isolation, detection, and interpretation of aerobic bacteriology cultures (Church, 2016). Blood specimens were cultured using the BACTEC™ automated system (Becton Dickinson, United States) and incubated at 35 °C for up to 3–5 days. Gram staining was performed on a positive blood culture, which was then sub-cultured into a solid medium. Skin microbiota (including coagulase-negative staphylococci (CoNS) and viridans streptococci) recovered from the culture were not reported as contaminants because we only collected one blood specimen (10 mL) per patient. Instead, we advised clinicians to match culture results to the patient’s infectious status before initiating antibiotic therapy. However, spore-producing Gram-positive rods grown in aerobic sheep blood agar (SBA) plates were most likely not reported.

Gram staining results determined the solid media used for wound and sputum culture. SBA plates were inoculated with 1 µL of clean midstream and urinary catheter specimens using a calibrated loop for colony count, then incubated in ambient air overnight at 35–37 °C. Gram-positive cocci were subcultured on SBA and mannitol salt agar plates. Gram-negative rods were subcultured on SBA, chocolate, MacConkey (MAC), and Eosin-Methylene Blue (EMB) agar plates (Oxoid™, Waltham, MA, UK). The cultures were incubated aerobically at 35–37 °C for 24–48 h. To recover yeast, specimens were inoculated on Sabouraud dextrose agar plate (Oxoid, Basingstoke, Hampshire, UK) and aerobically incubated at 37 °C for 48 h. We did not analyze sputum specimens for Mycobacterium tuberculosis because they were identified separately in an external reference laboratory (Adam Malik Hospital Clinical Microbiology Sub-installation laboratory).

Isolates of pathogenic bacteria and yeast grown in the culture media were subjects for species identification using the Vitek® 2 Compact commercial system with Gram-negative (GN), Gram-positive (GP), and yeast cards (bioMérieux, Marcy-l’Étoile, France). Antimicrobial susceptibility tests (ASTs) for bacterial isolates were done using the Vitek® 2 automated susceptibility system (bioMérieux, Marcy-l’Étoile, France) and Kirby-Bauer methods using antimicrobial susceptibility discs (Oxoid™, Waltham, MA, UK). Disk diffusion susceptibility testing for aerobic bacteria was performed by following the Clinical & Laboratory Standards Institute (CLSI) guidelines (CLSI, 2023).

Data analysis

The data were analyzed by descriptive statistics using Prism 10 (GraphPad Software, LLC., San Diego, CA, USA) and then presented in graphs and tables. Analysis of variance (ANOVA) was performed at a 95% confidence interval and p ≤ 0.05 using Prism 10. The antimicrobial susceptibility test (AST) results were shown only from the most isolated organisms, and the percentage of resistant organisms was reported in a table.

Results

Distribution of patients and clinical specimens

A total of 2,501 patients’ data were collected from January 2022 to December 2023, consisting of 1,292 (51.66%) males and 1,209 (48.34%) females. The distribution of patients based on age groups is presented in Table 1. Children were categorized in the age group below 18 (<18) as mentioned elsewhere (Ogbonna et al., 2019). The number of patients hospitalized in wards/inpatient [1,726 (69.01%)] outnumbered those admitted in the emergency department [522 (20.87%)] and the hospital clinics/outpatient [253 (10.12%)]. However, there are no significant differences in the number of patients between the age groups (one-way ANOVA, p = 0.175) and between the hospital admissions (one-way ANOVA, p = 0.071). During the 2 years, we collected 714 (28.54%) blood, 655 (26.18%) wound, 545 (21.79%) sputum, and 587 (23.49%) urine specimens. A total of 1,190 (47.58%) microbial isolates were recovered, consisting of 98 (13.73%), 454 (69.32%), 394 (72.29%), and 244 (41.57%) from blood, wound, sputum, and urine, respectively (Fig. 1).

Table 1 Demographics of patients.

	n (%)	P-value	
Male	Female	Total	
Gender	1,292 (51.66)	1,209 (48.34)	2,501	–	
Age	
<18	115 (4.60)	98 (3.92)	213 (8.52)	0.175	
18–24	68 (2.72)	56 (2.24)	124 (4.96)	
25–34	153 (6.12)	168 (6.72)	321 (12.83)	
35–44	158 (6.32)	157 (6.28)	315 (12.60)	
45–54	157 (6.28)	172 (6.88)	329 (13.15)	
55–64	316 (12.63)	252 (10.07)	568 (22.71)	
65–74	235 (9.40)	171 (6.84)	406 (16.23)	
≥75	90 (3.60)	135 (5.40)	225 (9.00)	
Hospital admission	
Emergency	283 (11.31)	239 (9.56)	522 (20.87)	0.071	
Inpatient	923 (36.90)	803 (32.11)	1,726 (69.01)	
Outpatient	98 (3.92)	155 (6.20)	253 (10.12)	
Specimens	
Blood	399 (15.95)	315 (12.60)	714 (28.54)	0.687	
Wound	365 (14.60)	290 (11.60)	655 (26.18)	
Sputum	326 (13.03)	219 (8.76)	545 (21.79)	
Urine	199 (7.95)	388 (15.51)	587 (23.49)	

Figure 1 Frequency of recovered isolates from specimens.

From the total of 2,501, we collected 714 blood, 655 wound, 545 sputum, and 587 urine specimens. Bacteria and yeast were recovered from 98 (13.73%), 454 (69.32%), 394 (72.29%), and 244 (41.57%) blood, wound, sputum, and urine specimens, respectively.

Distribution of microbial isolates

Following the microbial recovery from the aerobic culture, we identified bacteria and yeast, presented in Fig. 2. Gram-negatives (GNs) are the primary isolates [69/98 (70.41%)] recovered from blood specimens, pursued by the Gram-positives (GPs) [26/98 (26.53%)], and yeast Candida spp. [3/98 (3.06%)]. Escherichia coli was the most prevalent isolate identified from blood [27/98 (27.55%)] specimens. Previous studies reported E. coli as the most frequent GNs in patients with bloodstream infection, followed closely by Klebsiella pneumoniae (Sormani et al., 2023; McGuinness et al., 2024; Moon et al., 2024; Suganthini, Suvintheran & Nor Zanariah, 2024). E. coli from blood cultures indicates bacteremia, whether it is catheter-related (McGuinness et al., 2024) or the non-catheter related infections (Irigoyen-von-Sierakowski et al., 2024). Most patients with E. coli bacteremia in this study were from the age group of ≥ 55 years [17/27 (63%)] and thus associated with a higher risk of case fatality (MacKinnon et al., 2021). Coagulase-negative staphylococci (CoNS) [20/98 (20.41%)] were prevalent after E. coli in blood specimens, followed by Klebsiella spp. [16/98 (16.33%)], which was dominated by K. pneumoniae [15/16 (93.75%)]. Staphylococcus hominis [7/20 (35%)], S. epidermidis [3/20 (15%)], and S. haemolyticus [3/20 (15%)] were the three most common CoNS found in blood specimens.

Figure 2 The percentage of microbial isolates in different specimens.

(A) Isolates from blood specimens were dominated by E. coli (27.55%) and CoNS (20.41%). (B) E. coli (43.85%) was predominant in urine specimens, followed by K. pneumoniae (10.65%). (C) Sputum specimens, mostly recovered K. pneumoniae (17.26%) and Acinetobacter spp. (14.47%). (D) Wound isolates were dominated by E. coli (23.57%) and S. aureus (22.69%).

S. aureus was highly predominant [103/454 (22.69%)] in wound specimens after E. coli [107/454 (23.57%)], which was in accordance with former studies (Rajput, Johri & Goyal, 2023; Sheeba, Prathyusha & Anila, 2024). Frequent identification of E. coli and S. aureus in this study could be a clue of fecal contamination in wounds and nosocomial infection risk as an indication for further investigations regarding the proper management of wound infections. Similar to findings in blood and wounds, E. coli was also the most prevalent isolate in urine specimens [107/244 (43.85%)], followed by K. pneumoniae [26/244 (10.65%)]. E. coli isolates were found in 83 (77.57%) females and 24 (22.43%) males, while K. pneumoniae was found in 17 (65.38%) females and nine (34.61%) males with significant bacteriuria. Our results are in line with other studies (Khan et al., 2023; Hossain et al., 2024; McGuinness et al., 2024) confirming the role of E. coli and K. pneumoniae among uropathogenic bacteria.

K. pneumoniae [68/394 (17.26%)] predominated the microbial isolates from sputum, followed by Acinetobacter spp. [57/394 (14.47%)]. The latter was dominated by Acinetobacter baumannii [52/57 (91.22%)]. K. pneumoniae is consistently identified as the leading isolate in sputum specimens (Goel et al., 2022; Algurairy, 2024; Rafia et al., 2024), yet it is also saprophytic in the airway. Thus, K. pneumoniae’s presence in sputum samples does not always indicate a true pathogenic role and requires further examination to diagnose a lower respiratory tract infection.

Besides monomicrobial, we found polymicrobial isolates from 4/98 (4.08%) blood, 41/454 (9.03%) wound, 52/394 (13.2%) sputum, and 9/244 (3.69%) urine specimens. Pairs of GNs dominated the polymicrobial isolates from blood [2/4 (50%)], yet there was no GP pair. GP and GN pairs were prevalent in polymicrobial wound specimens [18/41 (43.9%)], while bacteria and yeast pairs dominated the sputum’s polymicrobial [26/52 (50%)]. GP pairs did not exist in urine specimens, yet the other pairs were found equally (Fig. 3A). The most common species pairs in polymicrobial isolates were K. pneumoniae and S. aureus [6/106 (5.7%)].

Figure 3 Characteristics of polymicrobial isolates by specimens.

(A) Prevalence of Gram-positives (GPs), Gram-negatives (GNs), Gram-positive and Gram-negative (GP + GN), and bacteria + yeast pairs in the polymicrobial isolates. A pair of GPs was not found in blood and urine specimens. (B) Prevalence of the non-MDR, non-MDR + MDR, and MDR pairs in polymicrobial isolates. There were no MDR pairs in blood specimens.

The polymicrobial pairs observed in this study were dominated by non-multidrug-resistant (non-MDR) strains (Fig. 3B). Pairs of non-MDR and MDR strains were found in 1/4 (25%) blood, 9/41 (21.95%) wound, 17/52 (32.69%) sputum, and 3/9 (33.33%) urine. At a lower percentage, pairs of MDRs were found in 4/41 (9.76%) wounds, 1/52 (1.92%) sputum, and 1/9 (11.11%) urine specimens. We found the highest percentage of MDR polymicrobial isolates in urine specimens, lower than the estimated prevalence in clinical settings (Harris et al., 2023). The most common MDR polymicrobial isolate in urine was extended-spectrum beta-lactamase (ESBL)-producing E. coli [4/5 (80%)]. A high prevalence of MDR strains was also found in polymicrobial isolates of sputum specimens, which were dominated by the Gram-positive CoNS [8/19 (42.10%)] and the Gram-negative ESBL-producing K. pneumoniae [4/19 (21.05%)].

Antimicrobial resistance profile of the bacterial isolates

Among Gram-positives, CoNS isolates from sputum showed the lowest sensitivity against most antibiotics tested (Table 2), followed by Enterococci from urine and wound specimens. S. haemolyticus was the dominant CoNS [17/28 (60.71%)] and the most common MDR CoNS isolate from sputum [15/28 (53.57%)]. S. epidermidis was the other common MDR CoNS in sputum [10/28 (35.71%)], which was dominated by methicillin-resistant S. epidermidis (MRSE) [6/10 (60%)]. E. faecalis was the most prevalent isolate [10/15 (66.67%)] among the uropathogenic Enterococci.

Table 2 Antimicrobial resistance profile of gram-positive isolates.

Source	Organism	N		Resistance (%)	
GM	FOX	CIP	LVX	VA	TGC	SXT	LZD	AM	DO	SYN	OX	CM	E	
Blood	S. aureus	3	33.3	66.6	33.3	33.3	0	0	0	0	–	0	0	50.0	0	0	
CoNS	18	38.8	50.0	16.6	35.3	0	11.1	41.2	11.7	69.2	15.4	12.5	62.5	66.6	62.5	
Wound	S. aureus	101	28.7	39.0	28.7	28.1	19.4	15.5	29.1	8.8	33.3	3.3	3.3	40.4	32.5	29.1	
CoNS	64	25	49.4	18.7	16.6	15.8	11.7	26.5	22.6	43.5	5.8	23.4	51.7	57.1	51	
S. pyogenes	3	33.3	–	–	0	0	0	33.3	0	0	0	0	0	0	33.3	
Enterococcus spp.	17	83.3	–	33.3	26.6	23.5	0	75	16.6	20	36.3	63.6	–	–	–	
Sputum	S. aureus	29	10.3	25.4	27.6	27.6	10.7	11.1	8.3	0	30.7	5.2	0	29.6	25	25.9	
CoNS	28	62.9	68.0	46.1	44.4	42.8	29.4	73.6	22.2	75	44.4	25	71.8	70.5	76	
S. pneumoniae	3	0	–	0	0	33.3	0	0	0	0	0	0	33.3	33.3	33.3	
Urine	S. aureus	9	11.1	0	0	0	0	0	0	0	66.6	0	0	0	0	0	
CoNS	19	31.5	54.5	33.3	31.5	15.7	11.1	31.3	17.6	33.3	18.2	20	76.5	68.7	50	
Enterococcus spp.	15	0	–	33.3	33.3	26.6	0	50	0	35.7	40	85.7	–	66.6	–	
Note:

N = the number of organisms tested against antimicrobial agents. Antimicrobial agents abbreviations: GM, Gentamicin; FOX, Cefoxitin; CIP, Ciprofloxacin; LVX, Levofloxacin; VA, Vancomycin; TGC, Tigecycline; SXT, Sulfamethoxazole-trimethoprim; LZD, Linezolid; AM, Ampicillin; DO, Doxycycline; SYN, Quinupristin-dalfopristin; OX, Oxacillin; CM, Clindamycin; E, Erythromycin. Non-reported results (−).

Resistance to cefoxitin and oxacillin was the most common among Gram-positives, dominated by CoNS (Table 2). Resistance to clindamycin and ampicillin came after, and it was also dominated by CoNS. Enterococci resistance to quinupristin-dalfopristin was high, as the majority of the isolates were E. faecalis [9/18 (50%) and 10/15 (66.67%) enterococci from wound and urine samples, respectively], which are mostly resistant to this antibiotic. Streptococcus pyogenes and Streptococcus pneumoniae, on the other hand, showed zero to minimal resistance to the antibiotics tested.

A. baumannii in wound specimens were the highly resistant isolates among the Gram-negatives (Table 3). Enterobacter cloacae was the sole species from Enterobacter found in urine [7/7 (100%)] and also showed a high level of resistance towards the majority of antimicrobials tested, followed by Pseudomonas aeruginosa isolates in wound and sputum specimens. Overall, the lowest sensitivity of Gram-negative isolates was shown towards ampicillin, followed by cefazolin, and ampicillin-sulbactam. K. pneumoniae isolates from urine and blood demonstrated increasing resistance to ceftazidime, a third-generation cephalosporin, most likely due to the beta-lactamase production such as ESBL and carbapenemase. Likewise, increasing resistance of E. coli isolates to ceftazidime indicated a significant proportion of the ESBL-producing strains.

Table 3 Antimicrobial resistance profile of gram-negative isolates.

Source	Organism	N	Resistance (%)	
GM	TZP	IPM	MEM	CZ	CAZ	FOX	CIP	SXT	TGC	ATM	AM	AMC	SAM	C	FOS	
Blood	E. coli	27	36.8	0	35.3	4.5	53.3	16.6	13	53.8	52.6	5.2	30.7	77.7	9.1	30.7	25	6.6	
Enterobacter spp.	7	42.8	25	33.3	28.5	83.3	44.4	50	57.1	71.4	14.2	57.1	100	85.7	100	33.3	0	
Klebsiella spp.	16	40	42.85	50	9	83.3	58.3	15.3	60	55.5	20	40	100	18.7	68.7	71.4	0	
Salmonella	5	100	0	0	0	60	0	0	50	0	0	0	20	33.3	20	0	0	
S. paucimobilis	5	40	33.3	25	25	66.6	60	75	20	20	0	60	50	50	50	0	66.6	
Wound	A. baumannii	19	57.1	61.1	45.4	53.3	89.4	68.4	78.5	45	30.7	0	83.3	81.8	77.7	47.3	87.5	54.5	
E. coli	102	28.5	13.8	42.4	7.6	61	27.6	15.7	35.3	52.1	0	34.3	78.5	14.4	48.5	45.7	11.1	
Enterobacter spp.	16	46.6	18.2	12.5	36.3	100	66.6	70	47	46.6	0	66.6	100	75	100	44.4	0	
K. pneumoniae	51	30.7	20.5	41.9	19	62.1	37.7	16.7	39.2	38.4	5.2	36	94.1	22.7	55.7	50	13.8	
Proteus spp.	8	20	37.5	75	0	100	16.6	100	50	60	100	25	50	16.6	50	0	0	
Pseudomonas spp.	28	23.1	31.8	33.3	37.5	100	48.2	75	48.2	78.5	88.8	55.5	75	82.6	100	87.5	34.7	
Sputum	A. baumannii	52	33.3	29.7	63.8	38.8	92.1	39.1	66.6	21.1	33.3	0	40	66	40.8	48.9	78.5	50	
E. coli	10	42.8	22.2	37.5	22.2	71.4	33.3	37.5	60	85.7	0	40	90	12.5	40	42.8	20	
Enterobacter spp.	14	27.2	25	40	25	100	42.8	60	35.7	38.4	11.1	38.4	92.8	58.3	92.8	42.8	16.6	
K. pneumoniae	68	27.1	24.1	36	11.7	46.5	27.5	21.8	35.8	25.5	8.3	30.3	91.1	23.2	52.3	48.5	23.5	
Pseudomonas spp.	34	15.6	33.3	47.3	37.5	90	30.3	70.8	34.2	100	90.6	51.4	72.7	79.4	85.7	81.2	47.8	
S. paucimobilis	28	16.6	11.7	15.8	13.8	23.8	14.3	21.7	24	33.3	8.3	50	17.6	16.6	14.2	12.5	35.3	
Urine	A. baumannii	7	50	0	57.1	20	85.7	25	28.5	0	50	33.3	50	57.1	0	75	0	0	
E. coli	105	25.3	13.7	43.5	6.5	53.7	29.3	8.7	44.8	61.3	0	36.2	82.8	15.3	50.4	48.1	5.7	
E. cloacae	7	66.6	50	20	16.6	83.3	83.3	100	85.7	83.3	16.6	71.4	100	83.3	100	50	0	
K. pneumoniae	26	48	34.7	21.4	31.5	63.6	68	33.3	76.9	68	36	73.1	100	45.4	69.2	72.2	25	
P. mirabilis	7	0	0	33.2	20	42.8	14.2	0	57.1	66.6	83.3	0	71.4	16.6	28.5	0	0	
P. aeruginosa	8	0	0	0	14.2	100	14.2	85.7	25	100	100	14.2	80	87.5	50	100	0	
Note:

N = the number of organisms tested against antimicrobial agents. Antimicrobial agents abbreviations: GM, Gentamicin; TZP, Piperacillin-tazobactam; IPM, Imipenem; MEM, Meropenem; CZ, Cefazolin; CAZ, Ceftazidime; FOX, Cefoxitin; CIP, Ciprofloxacin; SXT, Sulfamethoxazole-trimethoprim; TGC, Tigecycline; ATM, Aztreonam; AM, Ampicillin; AMC, Amoxicillin-clavulanic acid; SAM, Ampicillin-sulbactam; C, Chloramphenicol; FOS, Fosfomycin.

ESBL-producing E. coli was the most prevalent MDR found in patients with bacteremia, with equal distribution [5/49 (10.2%)] per year in 2022 and 2023 (Fig. 4). In 2022, ESBL-producing K. pneumoniae was less identified [1/49 (2.04%)] than methicillin-resistant S. aureus (MRSA) [2/49 (4.08%)]. However, in 2023, there was a fourfold increase in ESBL K. pneumoniae found in blood specimens [4/49 (8.16%)], surpassing MRSA [2/49 (4.08%)]. MRSA and ESBL E. coli both dominated the MDR isolates of wound specimens [9/206 (4.37%) each] in 2022, but MRSA increased [24/248 (9.68%)] more than ESBL E. coli [20/248 (8.06%)] in 2023. ESBL K. pneumoniae was the most common MDR isolate in sputum in 2022 [4/161 (2.48%)] and 2023 [12/233 (5.15%)]. MRSA was detected in sputum [7/233 (3.0%)] only in 2023. ESBL E. coli dominated the MDR isolates of uropathogens in 2022 [11/119 (9.24%)], and increased twofold in 2023 [22/125 (17.60%)].

Figure 4 Prevalence of multidrug-resistant bacteria by the year 2022/2023.

(A) MDR bacteria prevalence in blood specimens, dominated by ESBL-producing E. coli [5/49 (10.2%) per year]. (B) Prevalence of ESBL-producing E. coli was similar to MRSA in wound specimens in 2022 [9/206 (4.37%) each], yet MRSA increase was higher [24/248 (9.68%)] than ESBL E. coli [20/248 (8.06%)] in 2023. (C) MDR isolates in sputum were dominated by ESBL-producing K. pneumoniae 4/161 (2.48%) in 2022 and 12/233 (5.15%) in 2023. (D) ESBL E. coli was prevalent in urine specimens 11/119 (9.24%) in 2022 and 22/125 (17.60%) in 2023, followed by ESBL K. pneumoniae 5/119 (4.20%) in 2022 and 6/125 (4.80%) in 2023.

Discussion

Despite the highest number of blood specimens compared with other specimens obtained from the patients, it has the lowest positivity rate of microbial recovery from the culture (13.73%). A similar positivity rate from blood cultures was reported in previous studies (Neves et al., 2015; Lambregts et al., 2019; Patel, Patel & Patel, 2022). The low prevalence of bloodstream infections and administration of antimicrobial agents before specimen collection, notably the broad-spectrum antibiotics, reduces the positivity rate of blood culture (Johnson, 2019; Sari, Dahesihdewi & Sianipar, 2023). Other aspects that may reduce the positivity rate of blood culture are the non-standard blood culture collection patterns, including the blood volume, multiple and different puncture sites, and the use of anaerobic culture bottles along with aerobic culture bottles (Johnson, 2019; Wen et al., 2022).

We reported the highest positivity rate of aerobic culture from sputum specimens (72.29%) (Fig. 1). A high positivity rate of culture from sputum specimens supports better diagnostics, antibiotic therapy guidance, and clinical decisions regarding disease severity (Blakeborough & Watson, 2019; Crannage, 2022). However, positive cultures may also reflect colonization rather than active infection, and thus physicians must remain cautious of over-reliance on culture results as they can sometimes misrepresent the clinical picture, potentially leading to overtreatment.

Often considered a contaminant from skin surfaces (Bhosle, Thakar & Modak, 2022; Serra et al., 2023), the identification of CoNS species from blood specimens may lead to the suspected case of central line-associated bloodstream infection (CLABSI) (Haddadin, Annamaraju & Regunath, 2024). Two or more blood samples from different sites and the subsequent positive culture results are required to determine the clinical significance of CoNS isolates and confirm CLABSI in a suspected patient (Wright et al., 2018; Haddadin, Annamaraju & Regunath, 2024). The significant prevalence of S. haemolyticus and S. epidermidis among CoNS and MDR CoNS isolates in clinical specimens was mentioned elsewhere (Al Laham et al., 2017). The increasing CoNS and enterococci isolates represent a significant challenge in clinical settings, particularly in nosocomial infections (Bose et al., 2015; Goulart, 2023). Moreover, these pathogens exhibit high levels of resistance to multiple antibiotics (Bose et al., 2015; Mbanga, Sithabile & Silence, 2015; Al Laham et al., 2017; Khalil et al., 2023), complicating treatment options (Goulart, 2023). Increasing CoNS resistance towards beta-lactams in hospitals worldwide was mentioned elsewhere (Pereira et al., 2020; Goulart, 2023). Most hospital-acquired staphylococci produce beta-lactamase, and at least 60% are methicillin-resistant, making them resistant to the majority of beta-lactam antibiotics (Goulart, 2023). Furthermore, staphylococci strains that acquire mecA produce penicillin-binding protein 2a (PBP2a), which confers broad-spectrum resistance to beta-lactam antibiotics (Goulart, 2023).

Enterococci possess low-affinity penicillin-binding proteins, decreasing susceptibility to beta-lactams (Herrera-Hidalgo et al., 2023). Enterococcus faecalis strains carrying the lsaA gene are intrinsically resistant to quinupristin-dalfopristin (Sirichoat et al., 2020), an antimicrobial agent used to cure infection due to vancomycin-resistant Enterococcus faecium (VREF) (Reissier & Cattoir, 2020), posing significant challenges in determining its mechanisms and treatment options. Continuous monitoring of resistance patterns is crucial to inform treatment strategies and control the spread of these pathogens in healthcare settings.

The highest percentage of polymicrobial isolates was found in sputum (13.2%). Sputum culture tends to recover polymicrobial isolates due to the complex microbial communities in the respiratory tract, influenced by various factors such as underlying health conditions and sampling techniques (Grønseth et al., 2017; Crannage, 2022; Dickson, 2022). The incidence of polymicrobial isolates in this study is relatively lower than in previously reported studies (Mitchell, Yarbrough & Burnham, 2021; Hossain et al., 2024; Sheeba, Prathyusha & Anila, 2024). Polymicrobial isolates may reflect the true complexity of infections, suggesting that a broader approach to treatment may be necessary in certain cases. Moreover, studies revealed that biofilm formation due to the polymicrobial community in sites such as sputum and wounds can promote resistance and facilitate resistant gene transfer between bacteria, leading to treatment difficulties and longer hospital stays (Nabb et al., 2019; Chen et al., 2022; Kulshrestha & Gupta, 2022; Hamad & Alzubaidy, 2023). Further investigations are required to understand the complicated diagnosis as a consequence of polymicrobial infections.

Despite the opportunistic nature of E. cloacae, this bacterium is implicated in a range of infections, including urinary tract infections, pneumonia, and bacteremia, and accounts for 5–11% of nosocomial infections (Rizi, Ghazvini & Farsiani, 2020). Thus, the rising resistance levels of Enterobacter necessitate vigilant monitoring and effective infection control measures. The presence of efflux pumps and enzymes like beta-lactamases contributes to P. aeruginosa’s ability to resist multiple antibiotic classes (Kamel et al., 2024). Further, P. aeruginosa is known for its ability to form biofilms that protect it from antibiotic penetration, enhancing its antibiotic resistance (Gervasoni et al., 2023). Apart from the limitations of the retrospective nature of the study methods, our results indicate high resistance of Gram-negatives against beta-lactams, including cephalosporins. Because this study relied on hospital e-medical records, any missing data could aggravate recall bias in reporting the results. Qualitative research may address specific questions that were not addressed in this study.

The alarming resistance patterns of A. baumannii from clinical specimens were frequently mentioned (Mayasari & Siregar, 2014; Zidan, Samanje & Nasir, 2022; Flaifel, 2023), indicating the need for ongoing surveillance and innovative therapeutic approaches. Gram-negative bacteria produce enzymes such as ESBLs and carbapenemases that hydrolyze cephalosporins, rendering them ineffective (Pfeifer, Cullik & Witte, 2010). As a barrier, the outer membrane of Gram-negative can limit the penetration of cephalosporins, which is exacerbated by mutations that reduce porin channel expression (Xu et al., 2022). Further, overexpression of efflux pumps can actively expel antibiotics from the bacterial cell, contributing to resistance (Xu et al., 2022; Gaurav et al., 2023). Despite the intense use of carbapenems, we found a low level of Gram-negative resistance against these antibiotics. However, the decreasing level of sensitivity against imipenem (Table 3) shall be a clinical concern.

Overall, our results revealed the prominence of ESBL E. coli and ESBL K. pneumoniae among the MDR isolates, with an increasing number from the year 2022 to 2023. The presence of ESBLs often necessitates the use of carbapenems, especially in patients with bacteremia, where inappropriate initial therapy can lead to poor outcomes (Guaraná et al., 2022). While carbapenems are often considered the treatment of choice for infections caused by ESBL-producing bacteria (Guaraná et al., 2022; Maseda & de la Rica, 2022), indiscriminate use of these antibiotics can contribute to the global antimicrobial resistance crisis (Tamma & Mathers, 2021). Alternatives including piperacillin-tazobactam and newer beta-lactam/beta-lactamase inhibitors, such as ceftazidime-avibactam, are being explored to reduce reliance on carbapenems (Maseda & de la Rica, 2022). The increasing trend of ESBL-producing Gram-negative presented in this study implicates the need for continuous surveillance, effective infection control, and appropriate treatment protocols.

Conclusions

The Gram-positives S. aureus and CoNS, and the Gram-negatives E. coli and K. pneumoniae were predominant bacterial isolates from blood, wound, sputum, and urine specimens in this study. Corresponding to those findings, we also identified ESBL E. coli and ESBL K. pneumoniae as the most prevalent MDR isolates. Resistance to cefoxitin and oxacillin was the highest among the Gram-positives, followed by increasing resistance against clindamycin and ampicillin. In Gram-negatives, resistance against ampicillin was the highest, and subsequently, the resistance against cefazolin and ampicillin-sulbactam. This epidemiological study on microorganism infections must be updated periodically, especially in a hospital setting, as it is crucial to develop the prevention and infection control program and the updated empirical antibiotics in the hospital.

Supplemental Information

Supplemental Information 1 The STROBE Checklist for a cross-sectional study.

We acknowledge the creators of SciSpace for a fast-writing experience.

Additional Information and Declarations

Competing Interests

The authors declare that they have no competing interests.

Author Contributions

Evita Mayasari conceived and designed the experiments, performed the experiments, analyzed the data, prepared figures and/or tables, authored or reviewed drafts of the article, and approved the final draft.

Edhie Djohan Utama conceived and designed the experiments, performed the experiments, analyzed the data, authored or reviewed drafts of the article, and approved the final draft.

Ethics

The following information was supplied relating to ethical approvals (i.e., approving body and any reference numbers):

This study was approved by the Ethics Committee of the Universitas Sumatera Utara as registered in the letter number 1000/KEPK/USU/2023, and the Hospital’s Ethics Committee in the letter number 1873/SHMD/DIR/SB/XII/2024.

Data Availability

The following information was supplied regarding data availability:

The data is available at Zenodo: Mayasari, E. (2025). evita-eng/retrospective_study2025: raw_data_peerJ2025 (author). Zenodo. https://doi.org/10.5281/zenodo.14920876.

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
