# Peer review of "Epidemiology and resistant profile of bacterial pathogens in a tertiary health care hospital, Medan City: a retrospective study"

_PeerJ, doi:10.7717/peerj.19510_

## Round 0.1 · original submission · Major Revisions

Dear authors, thank you for your submission. At the moment, your manuscript still do require significant revisions in order to be acceptable for publication. Please, refer to the reviewers' comments for further details.

Reviewer 1 ·

Basic reporting

The review manuscript entitled "Epidemiology and resistant profile of bacterial pathogens in a tertiary health care hospital, Medan city" is an interesting topic. Generally, the overall text and contents are clear and unambiguous but it still needs professional editing throughout. Literature references are sufficient and up to date.

For background/context provided especially in Introduction, there comments that authors can modify accordingly:
1st paragraph...
Overview of Infectious diseases globally
2nd paragraph
Epidemiology of drug resistant
3rd paragraph touch on the role of diagnostic and preventive measures
4th paragraph
In Indonesia....what happens?.
There are no proper Objectives of this study, therefore, a list of Objective(s) can be added at the last part of this Introduction.
So that description of the Results and detailed information of Discussion can be in consistent with the objectives. This will make it more scientific values and better understanding.

For cited references, authors can choose between citing from early to recent years of references OR vice versa.

In full references, I have made some changes to keep the standard format throughout (Refer to the attached file)

Tables 1 and 2 should be revised because it is so plain with those demographic and frequency. It will be more meaningful with comparison between groups and p-values, as appropriate so that it will be more robust and interesting that researchers/readers can use it for further study and references in the future. Please add p-values at the Table footnote for the comparative study. Tables 3 and 4 might as well try for any comparative analysis (if statistical results, add at the Tables footnotes) for more convincing data and results from this study and that can be used for references in future studies.

Experimental design

The overall study falls into the aims and scope of the journal. In addition, authors can add more information related research that can fill an identified knowledge gap.

For Methodology, there are comments that authors can modify to more rigorous, sufficient detail and information to replicate in the future.
1. Authors can provide the overall Methodology as an infographic figure to enhance scientific values and better understanding.
2. It would better if authors can provide proper inclusion and exclusion criteria of this study.
3. All chemicals, instruments and diagnostic kits should be cited with (company, city, country) throughout.
4. Authors can provide full term (in short) as the first mentioned wording in the text.
5. Lines 88-9, authors can provide a brief of these protocol techniques for better understanding.
Please go through all detailed information in this Section again for a better version of this study.

Validity of the findings

In Discussion, authors can add more pros and cons in the limitation of the study for more meaningful replication in future study.

All data provided especially in Table formats; it would be better to also provide statistical analysis for any comparison found in the study.

For Conclusion, it is generally well stated, linked to original research question and limited to supporting results.

Additional comments

More comments can be found in the attached file.

Annotated reviews are not available for download in order to protect the identity of reviewers who chose to remain anonymous.

·

Basic reporting

The Manuscript “Epidemiology and resistant profile of bacterial pathogens in a tertiary health care hospital, Medan city is in accordance with the guidelines of the journal, еthically sound.
Many errors were found and some parts of the manuscript could be improved.
Recommendations:
1. The manuscript lacks criteria for significance of the isolate with an etiologic agent. They must be defined.
2. Were some quantitative criteria used according to inoculation methods?
3. Streptococcus spp. It consists of many different streptococci. Are they identified by species? Some of them are pathogenic, others are part of the normal microbiota of mucous membranes of upper respiratory tract and other ecological objects of the human body. List them as different species.
4. The phrase "dominated by Enterobacter spp" should be deleted in the sentence on page 2, line 35-36 “In Gram-negative, susceptibility to cefazolin is the lowest, dominated by Enterobacter spp.” This species has innate resistance to all first-generation cephalosporins.”
5. Staphylococcal resistance to carbapenems should match methicillin resistance. These results for imipenem and meropenem are inaccurate and should be eliminated from the table 2. Only resistance to oxacillin or cefoxitin should be present.
6. Enterococci should be interpreted as resistant to macrolides and oxacillin and do not need to be present in Table 2 test results to these groups of antibiotics
7. Streptomycin (STS) presented below after the table 3, but there is no place for this antibiotic here.
8. Enterobacter spp., A. baumannii, P. aeruginosa should be interpreted as resistant to aminopenicillins alone and including plus a beta-lactamase inhibitor, also first generation cephalosporins, fosfomycin and there is no need to comment on their results.
9. The discussion is very poor. Something should be discussed about the resistance mechanisms that are typical of a particular group of bacteria.
10. Nothing is discussed about streptococci resistance. This part needs to be developed.

Experimental design

The methods are old routine.

Validity of the findings

The discussion is very poor. Something should be discussed about the resistance mechanisms that are typical of a particular group of bacteria.
Nothing is discussed about streptococci resistance. This part needs to be developed.

---

## Round 0.2 · Minor Revisions

Dear authors,
Please, proceed with the minor revisions as per the reviewer suggestion. Please, do not forget to proofread and make sure that all figure and table legends are complete and with enough information.

Reviewer 1 ·

Basic reporting

The revised version is acceptable based on the overall data provided in this study.

Experimental design

NA

Validity of the findings

NA

Additional comments

There are a few more corrections that need to be revised in the attached file.

Annotated reviews are not available for download in order to protect the identity of reviewers who chose to remain anonymous.

---

## Round 0.3 · accepted · Accept

Dear authors,
I am now accepting your manuscript for publication as all issues raised by the reviewers have been answered. Congratulations!